# Magnetoimpedance of CoFeCrSiB Ribbon-Based Sensitive Element with FeNi Covering: Experiment and Modeling

**DOI:** 10.3390/s21206728

**Published:** 2021-10-10

**Authors:** Stanislav O. Volchkov, Anna A. Pasynkova, Michael S. Derevyanko, Dmitry A. Bukreev, Nikita V. Kozlov, Andrey V. Svalov, Alexander V. Semirov

**Affiliations:** 1Department of Magnetism and Magnetic Nanomaterials, Institute of Natural Sciences and Mathematics, Ural Federal University, 620002 Ekaterinburg, Russia; stanislav.volchkov@urfu.ru (S.O.V.); nikita.kozlov@urfu.ru (N.V.K.); andrey.svalov@urfu.ru (A.V.S.); 2Laboratory of Advanced Magnetic Materials, Institute of Metal Physics UD RAS, 620108 Ekaterinburg, Russia; 3Department of Physics, Pedagogical Institute, Irkutsk State University, 664003 Irkutsk, Russia; mr.derevyanko@gmail.com (M.S.D.); da.bukreev@gmail.com (D.A.B.); semirov@mail.ru (A.V.S.)

**Keywords:** magnetic field sensors, rapidly quenched amorphous ribbons, thin films, magnetic composites, computer simulation, finite elements method, thin film, magnetic field sensors

## Abstract

Soft magnetic materials are widely requested in electronic and biomedical applications. Co-based amorphous ribbons are materials which combine high value of the magnetoimpedance effect (MI), high sensitivity with respect to the applied magnetic field, good corrosion stability in aggressive environments, and reasonably low price. Functional properties of ribbon-based sensitive elements can be modified by deposition of additional magnetic and non-ferromagnetic layers with required conductivity. Such layers can play different roles. In the case of magnetic biosensors for magnetic label detection, they can provide the best conditions for self-assembling processes in biological experiments. In this work, magnetic properties and MI effect were studied for the cases of rapidly quenched Co_67_Fe_3_Cr_3_Si_15_B_12_ amorphous ribbons and magnetic Fe_20_Ni_80_/Co_67_Fe_3_Cr_3_Si_15_B_12_/Fe_20_Ni_80_ composites obtained by deposition of Fe_20_Ni_80_ 1 μm thick films onto both sides of the ribbons by magnetron sputtering technique. Their comparative analysis was used for finite element computer simulations of MI responses with different types of magnetic and conductive coatings. The obtained results can be useful for the design of MI sensor development, including MI biosensors for magnetic label detection.

## 1. Introduction

Magnetic materials are widely used in electronics and biomedical applications [1,2]. There are different types of ferromagnets designed for existing and proposed technological applications in in different fields, including medicine [3,4]. Among others, magnetic materials for detectors of small magnetic fields were considered in this research area [5,6]. The magnetic effect that ensures the highest sensitivity with respect to an applied magnetic field is the magnetoimpedance (MI) [7,8,9]. It can be used both for detection of the biomagnetic signals closely related to the functional activities of the living systems and magnetic label detection [3,6,10,11]. Special attention was paid to the development of the devices for the detection of the nanoparticles inside the living cells, in the blood stream or incorporated into a natural tissue [3,12,13]. The first prototype of MI biosensor for the detection of magnetic nanoparticles of iron oxide in commercial water-based stable suspension (ferrofluid) employed a Co-based amorphous ribbon as a magnetic sensitive element [14]. For the magnetic biosensing of the magnetizable labels, flat geometry is crucial because the biochemistry step includes various processes of self-assembling and washing.

Nanoparticles were not immobilized at the surface of the sensitive element. As the biomedical applications request nanoparticles provided as ferrofluid, all model experiments with ferrofluid detection have an additional value. Latter, this kind of testing was used for characterization of the properties of different kind of stable suspensions [15]. Co-based amorphous ribbon is a well-known material with a very low negative value of the magnetostriction coefficient, low coercivity, high magnetic permeability [5,6,7,8,9,10], and also high temperature stability due to chromium additions [16,17]. One of the most interesting and well-understood features is the possibility of creation of the induced magnetic anisotropy of the desired type [7,18].

The MI effect is associated with the high magnetic softness of the ferromagnetic conductor and the possibility to create a well-defined uniaxial magnetic anisotropy in the MI sensitive element [18]. The magnetoimpedance phenomenon consists of a change of the complex electrical impedance of ferromagnetic conductor under an influence of an external magnetic field [7,19,20]. The MI value in the amorphous cobalt-based ribbons and wires can reach hundreds of percent in weak magnetic fields of the order of a few Oersted [7,8,9], which distinguishes it favorably from other magnetic effects [21,22]. Therefore, their application in the area of the magnetic sensors [6,9], adapted for biodetection, is actively discussed, and proven in different compact analytical devices prototypes [11,14].

MI can be described in terms of the classic skin-effect [7,8,9,20], which consists of the inhomogeneous distribution of the current density over the cross section of the conductor. Due to the skin-effect, the density of the alternating current decreases in the direction from the surface toward the central part of the conductor. The skin-effect can be characterized by the penetration depth (*δ*) or the skin-depth:(1)$$\delta =c/{\left(\pi f\sigma \mu_{t}\right)}^{\frac{1}{2}},$$
where *σ*—the conductivity, *f*—the alternating current (AC) frequency, *µ_t_*—the dynamic effective transverse magnetic permeability. It can be seen that the greater the magnetic permeability, electrical conductivity, and AC frequency, the smaller the depth of the skin layer and the more pronounced the skin-effect. The stronger the skin-effect, the stronger the difference between the effective and geometric cross-section of the conductor. Despite the fact that these representations are quite straight forward, a simple derivation of analytical expressions for the impedance is possible only for a limited number of idealized symmetric homogeneous cases [7,14,15]. Ref. [14] describes the first MI-based prototype of the biosensor for magnetic label detection. The last case not only requires sufficient sensitivity of the prototype with respect to applied field, but the flat geometry is also very important as the biofunctionalization and other related processes take place in a liquid and require manipulation of the biofluids.

One of the main tasks of MI-sensor development is optimization of their MI parameters in accordance with particular application request. In the case of the homogeneous MI element, the main task is the corresponding optimization of its magnetic properties [7,19,20,23,24]. Much wider possibilities for the optimization of MI responses are available in the case of the multilayered structures. Variations of the geometric parameters are also challenging. For example, in comparison with a continuous film, much higher MI values were achieved in multilayered structures, in which a conductive layer is located between layers with a high magnetic permeability [25,26].

More complex layered structures were also proposed and investigated. For example, a new type of MI structures has been proposed and studied theoretically for the case of a multilayered structure consisting of a highly conductive central layer and two outer ferromagnetic layers below and above the conductive one. The upper layer is a periodic structure, it consists of N multilayer elements and N + 1 regions in which there are no multilayer elements (the upper layer is profiled). An electrodynamic model has been created that allows one to find the values of the transverse magnetic permeability for the upper and lower layers of the MI structure. It is shown that for a profiled structure with a decrease in the deviation angle of the effective magnetic anisotropy axis from the transverse direction, the magnetic permeability of the upper layer increases, which leads to an increase in the skin effect and an increase in the MI effect [27].

Permalloy layers obtained by magnetron sputtering have been repeatedly investigated by the scientific community. Their composition and magnetic anisotropy can be very well controlled in stable deposition conditions [28,29,30]. FeNi-based MI prototypes were designed and developed [26,27,31,32].

On the other hand, the tendency of the development of the functional multilayered structures (for example, coated amorphous ribbons) is dictated by the goal of the functionalization of the surface of the MI element. So, it seems expedient to use gold, magnetite (Fe_3_O_4_), reduced graphene oxide (rGO), or iron coatings [33,34,35,36]. They are widely adapted in the self-assembling processes in biotechnologies in the tasks of the marker biodetection [5,11]. This area has recently become a very hot topic, especially when focused on carbon containing nanostructures with enhance functional properties [34,35].

An optimization of the MI response of a layered structure involves varying the material of the layers, induced magnetic anisotropy features and the geometric dimensions. A direct experimental search for a solution to this problem is very laborious, and obtaining of the analytical expressions is much more difficult than in the case of homogeneous sensitive elements. Therefore, the finite element method (FEM) has recently been used by researchers in the field of MI [37,38,39]. This approach allows us both to find the optimum configuration of the MI element and to simulate its operation in various conditions. A lot of work has been done on modeling the MI response in thin film structures [12,24,39] including those adapted for biodetection [40]. However, there are almost no studies on modeling the MI response of amorphous ribbons particularly for specialized configurations [41,42]. This applies to an even greater extent to the amorphous ribbons with coatings that functionalize their surface. However, the use of amorphous ribbons for biodetection is actively discussed (for both label-free and magnetic label detection) [4,11,15]. For this reason, the development of the computer models for the estimation of the properties of MI elements based on amorphous ribbons seems to be an important task.

This work presents the results of comparative studies of the magnetoimpedance effect of the amorphous CoFeCrSiB ribbons both with and without the FeNi magnetic layer coating. The experimental data for particular conditions are used for extended FEM analysis and prediction of the MI behavior of complex composites. In addition, the results of computer simulation are presented for MI ribbon-based materials with other coatings (Cu, Au, Fe, Fe_3_O_4_), which could be used in order to functionalize their surface in real biomedical devices.

## 2. Materials and Methods

### 2.1. Samples and Experimental Methods

Amorphous Co_67_Fe_3_Cr_3_Si_15_B_12_ ribbons (IR—ribbons in the initial state) were prepared by the rapid quenching from the melt onto the rotating copper weal technique. Quenching then proceeded in the air. The thickness and the width of the ribbons were 20 μm and 2 mm respectively. The following composition of amorphous ribbons was selected on the basis of previous studies at the same laboratory [43]. All previously defined structural and magnetic parameters were checked by corresponding techniques and their values were confirmed with the accuracy above the 5%: the ribbons had the crystallization temperature 570 °C and saturation magnetostriction λ ≈ −0.2 × 10^−6^.

The FeNi/CoFeCrSiB/FeNi composites (R-FeNi) were obtained by the deposition of Fe_20_Ni_80_ thin films at the bottom and on the top planes of the IR ribbon. The deposition was done at room temperature of the substrate by the magnetron sputtering method in Ar-atmosphere on ATC Orion Series Sputtering Systems (AJA International, North Scituate, MA, USA). The following parameters were applied: the background pressure was 3.0 × 10^−7^ mbar and working argon pressure 3.8 × 10^−3^ mbar. For every deposition onto the ribbon surface, we add additional piece of the substrate in the area of uniform deposition and the composition was specially checked afterwards by energy dispersive X-ray analysis technique: the permalloy film composition was very close to Fe_20_Ni_80_ corresponding to zero magnetostriction with an accuracy of about 1%. More details on the deposition conditions can be found elsewhere [4].

The thickness of the FeNi layers was 1 μm on each plane of the ribbon sample (Figure 1a).

The optical microscopy photographs showed the surface features typical for amorphous ribbons of this composition. The most important point is that the FeNi deposited layer formed rough but continuous layers in both sides of the IR ribbons (Figure 1b). The observed roughness of the FeNi layer corresponded well to the features of the roughness of the ribbon itself. X-ray diffraction analysis of the IR was performed using a X’PERT PRO diffractometer (Philips, MX Amsterdam, The Netherlands) in Cu-Kα radiation.

Magnetic hysteresis loops of the IR samples were measured by the induction method by applying an external quasi-static magnetic field along the length of the sample (in-plane configuration). In addition, the amorphous ribbons were studied by the magneto-optical Kerr (MOKE) microscope and magnetometer (Evico magnetics GmbH, Dresden, Germany).

The magnetoimpedance measurements were carried out using the automatic system based on the Agilent impedance analyzer 4294A (Agilent/Keysight Technologies, Santa Rosa, CA, USA) using all necessary calibrations in order to extract the intrinsic impedance value corresponding to the signal of the magnetic sensitive element [9,42,43]. We studied the frequency range of an alternating current, *f*, of 0.1 to 70 MHz with an effective current value of 10 mA. The external magnetic field, *H*, was oriented along the long side of the sample, parallel to the direction of the flow of the alternating current, i.e., longitudinal MI configuration was employed. The maximum intensity of the external magnetic field, *H_max_*, was as high as 150 Oe. The MI effect ratio was calculated as follows:(2)$$\Delta Z/Z(H)=\frac{Z(H)-Z(H_{max})}{Z(H_{max})}\times 100\%,$$
where *Z*(*H*) and *Z*(*H_max_*) are the impedance modules in the magnetic fields *H* and *H_max_*, respectively.

The magnetic field sensitivity of the impedance was determined by the expression:(3)$$S=\frac{\partial Z}{\partial H},$$
where *∂**Z*—is the impedance difference per magnetic field change discrete *∂**H*.

The samples for magnetic and MI measurements were 50 mm long.

### 2.2. Computer Simulations

The magnetodynamic behavior of the ribbons with and without covering was preliminary modeled using the computer simulation by the finite element method, which allows us to obtain a numerical solution of differential non-linear equations for a magnetoimpedance effect taking into account the geometry and physical-chemical parameters of the magnetic system [44]. The simulation was carried out using the Comsol Multiphysics 5.6 licensed software -core and AC/DC module (COMSOL AB, Stockholm, Sweden).

Typically, the shape and size of a finite element is determined by the geometrical parameters of the system in the conditions in which physical properties can be changed (for example, the presence of roughness, the size of the inhomogeneities of magnetic properties, etc.). In solving this problem, a tetrahedral non-structured mesh (network with uneven coupling) was used, since there is a large number of domains where the magnetic properties of the element are anisotropic. The solution uses a tetrahedral grid generator based on Delaunay algorithms [45].

The model of the amorphous CoFeCrSiB ribbon for computer simulation took into account the parallelogram geometry (coinciding with the shape of the sensitive element) (Figure 1a). Using software, the model was divided into tetrahedral sub-domains for 3D configuration. The size of the finite elements in the partition mesh depends on the wavelength of the electromagnetic excitation. A separate subsection in the form of hexahedral elements was established for the near-surface layer of the element (~20 nm), due to the prevailing current distribution in the condition of the significant skin-effect. The model with thin covering layers on top and at the bottom planes (Figure 1a) takes into account the program function of transient boundary conditions due to the presence of diffusion of the order of several nanometers between the covering and the ribbon and different propagation value of the electromagnetic wave in layers with different conductivity and magnetic permeability.

At the frequencies of alternating current above 1 MHz, the magnetization process is carried out only by the rotation of the magnetization vector. Therefore, the displacements of the magnetic domain boundaries cannot be taken into account as they are dumped by the eddy currents [7]. Therefore, the expression for the effective transverse magnetic permeability can be obtained using the procedure for minimizing the free energy functional, as described, for example, in [17]:(4)$$\mu_{t}=1+\frac{M_{S}\sin^{2}\theta }{H\sin\theta +H_{K}\cos2\left(\theta -\psi \right)},$$
where *M*_*S*_ is the saturation magnetization, *H*_*K*_ is the transverse magnetic anisotropy field, *ψ* is the angle between the effective anisotropy axis and the transverse direction and *θ* is the angle between the magnetization and the direction of the external magnetic field *H*. In this case, the angle *θ* is related to *H* by the following expression:(5)$$H_{K}\sin(\theta -\psi )\cos(\theta -\psi )=H\cos\theta .$$

The values of *M*_*S*_, *H_K_* and *ψ* were set separately for each layer of the FEM model based on the results of work [19].

The models of the CoFeCrSiB ribbons with the following coatings were developed: FeNi, Cu, Au, Fe, Fe_3_O_4_. Copper and gold-based coatings have been selected as frequently used high conductivity materials. The designations for the samples analyzed in different models, as well as some properties of the coatings are given in Table 1. The thickness of the coatings in the FEM models varied from 10 to 1500 nm, which is the reasonable interval for continuous layers.

## 3. Results and Discussion

### 3.1. Experimental

According to X-ray analysis, the IR samples were amorphous. There was only an increase of intensity between the 2*θ** angles 40°–55° corresponding to amorphous halo. No bright peaks corresponding to crystalline structures were observed.

Magnetometry studies in quasi-static conditions for the ribbon samples without and with FeNi covering have shown that only the magnetic saturation field and the saturation magnetization were slightly changed after FeNi covering of the initial CoFeCrSiB ribbon. The remnant magnetization and coercive force were unchanged (Figure 2a). Based on the shape of the magnetic hysteresis loops, it can be concluded that the effective magnetic anisotropy of both IR and R-FeNi samples is rather complex. *M*(*H*) dependence is not linear for in-plane magnetization. On one hand, the coercivity is very small and magnetic hysteresis is almost negligible: *M*(*H*) curves in increasing and decreasing magnetic fields are very close to each other. However, the slope is also small, and the saturation field is of the order of 1 Oe. Similar behavior was previously observed in Co-based rapidly quenched amorphous ribbons without additional heat treatments [46,47].

For the description of the effective magnetic anisotropy, one should consider the main contribution to be a longitudinal magnetic anisotropy with the anisotropy axis oriented along the long side of the ribbon. Figure 2 describes the surface magnetic domains revealed by the MOKE microscopy under applications of the external magnetic field in the plane of the sample along the long side of the ribbon starting with high positive magnetic field in the sequence order: (b)–(c)–(d)–(e)–(f).

One can see that magnetization process is indeed quite complex but expected for Co-based rapidly quenched amorphous ribbons without additional heat treatments [39]. It is very probable that surface domains reflect the domain structure inside the sample in a very rough way and that the surface anisotropy contribution is quite high. Magneto-optical studies of the FeNi layer of the R-FeNi sample showed that the shape of the magnetic hysteresis loops corresponds to the so-called “transcritical” state, due to the occurrence of anisotropy of the perpendicular surface [6,22,48]. It is associated with the formation of perpendicular magnetic anisotropy component, formation microstructure columnar during sputtering of the FeNi coating deposition, and the appearance of the stripe domains. Therefore, magnetron sputtering was chosen as a method for obtaining nanocrystalline permalloy with stable structural characteristics and well-defined magnetic anisotropy. However, different deposition techniques such as electrodeposition can also be effective [49,50].

As the volume of the permalloy film is small in comparison with the volume of the amorphous ribbon in the R-FeNi composite, parameters such as saturation magnetization, remanence magnetization and magnetic coercivity of the composite are close to the parameters of the ribbon.

Magnetoimpedance dependences for both IR and R-FeNi samples again have a complex shape. On one hand, the curve tends to approach a “two peaks” shape (Figure 3a) with two maxima near the field close to magnetic anisotropy fields for positive and negative *H* values. At the same time the Δ*Z*/*Z*(*H* = 0 Oe) ≠ Δ*Z*/*Z*(*H* = 150 Oe) and the observed difference is quite significant. This means the existence of two strong contributions to the effective magnetic anisotropy: the longitudinal (most probably corresponding to the bulk part of the ribbon [51,52]) and the transverse (related to the surface). This, like the hysteresis loops (Figure 2a), indicates predominantly longitudinal magnetic anisotropy but with very strong contribution of the transverse component [19,52]. In the case of the IR sample, the increase of the frequency from 1 to 10 MHz results in the significant decrease of the Δ*Z*/*Z* ratio and the appearance of the Δ*Z*/*Z* maximum in the lower external magnetic field. This is consistent with the supposition that the transverse anisotropy component is mostly associated with the surface anisotropy and a fairly large local anisotropy axis distribution near the surface. The last supposition is confirmed by the magnetic domains observations in the IR sample.

The MI of FeNi-coated ribbons is noticeably lower (Figure 3) than the initial ribbons in the entire frequency range. In part, this is due to a decrease in the transverse effective magnetic permeability due to the “transcritical” state of the FeNi coating.

However, the shape of the Δ*Z*/*Z*(*H*) curve for f = 1 MHz is much closer to the “one peak” shape corresponding to the longitudinal effective anisotropy (Δ*Z*/*Z*(*H* = 0 Oe) ~ Δ*Z*/*Z**max*) with small surface-related peak height anisotropy [53]. As the sputtered FeNi layer due to the shading effect has non-uniform thickness but at a time reduces the surface roughness and insures the better closing of a magnetic flux near the surface favoring the longitudinal anisotropy contribution. The effect of the coating is increased with the increase in the alternating current frequency, due to the skin-effect (see Expression (1)). So, at a frequency of 10 MHz and above, there is a sharp decrease in the MI of the R-FeNi samples, as well as a significant shift of the maximum to the region of large values of the magnetic fields. This phenomenon can be explained by the flow of alternating current mainly in the permalloy layer due to the skin effect. The presence of maxima of the Δ*Z*/*Z*(*H*) curves in the case of composites also indicates the presence of a transverse magnetic anisotropy component but the main decay of the Δ*Z*/*Z* ratio at high frequencies can be due to the flow of the alternating current mainly over the region with very high inhomogeneities in the structure (Inset Figure 3a).

The magnetic field sensitivity of the impedance of both IR and R-FeNi structures, calculated using Expression (3), reaches a maximum, *S_max_*, in the range of the magnetic fields from 0 to *H*, corresponding to (Δ*Z*/*Z*)*_max_*. Frequency dependences of *S_max_* are shown in Figure 3b. It can be seen that *S_max_* changes non-monotonically with AC frequency increase. In the case of uncoated ribbons, the highest sensitivity of about 5 Ohm/Oe was observed at the frequency of 20 MHz. For the coated ribbons, the highest sensitivity was about 0.3 Ohm/Oe and was observed at the frequency of 30 MHz. The sensitivity of the MI of the R-FeNi samples with respect to the applied magnetic field is an order of magnitude lower than the sensitivity of the IR samples in the entire investigated frequency range. However, the obtained result can be used for estimation of the behavior of different composites with variation of the type and the thickness of the covering. Such a usage of modern computer technologies is very useful for complex and time-consuming technological processes.

### 3.2. Computer Simulation

Figure 4 shows the results of the computer simulation of the MI responses of IR and R-FeNi ribbons. It is very important to mention that the values of *M*_*S*_ = 450 Gs, *H_K_ =* 0.75 Oe, and *σ* (Table 1) required for FEM modeling were determined from experimental data obtained in the present study (*ψ =* 0.1 rad). The choice of the angle of anisotropy was based on past parametric studies [19,44] in such a way that the maximum magnetic permeability of a sample with a transverse uniaxial magnetic anisotropy was achieved at a nonzero value of the linear magnetization component. The simulated *Z*(*H*) dependences have the same character as the experimental ones for the uncoated ribbons. For the AC frequencies below 10 MHz the experimental and simulated curves are very close to each other. In the case of the case of coated ribbons, their character repeats the character of the experimental dependences only in the current frequency range of 1 MHz and below. However, it was possible to reproduce the main experimental result for given thickness of the covering with the help of simulation. MI significantly decreases after the FeNi layer deposition onto both surfaces of the amorphous CoFeCrSiB ribbon.

The main observations for the obtained experimental and calculated dependences with an increase in the AC frequency can be summarized as follows:(a)The proposed FEM model is very simple, and it does not take into account the frequency dependence of the magnetic permeability, which can be a rather complex function for the frequency range under consideration (see Expressions (4) and (5)) [51].(b)The non-flat morphology of the ribbon’s surface is not considered (Figure 1b and Figure 3a). Irregularities on the surface lead to a dispersion of the anisotropy [52,54], i.e., *ψ* (see Expressions (4) and (5)) takes on different values at different points of the ribbon’s surface.(c)The dispersion of the anisotropy over the cross section of the ribbon is not taken into account [52,54,55].(d)The magnetic interaction peculiarities, including the features related to the variation of the additional layer thickness, for the amorphous ribbon and the FeNi coating are not considered.

Nevertheless, even the presented FEM solved model reproduces the experimental results at a most simple qualitative level.

Keeping in mind the fact that surface properties of the ribbons can be adjusted by the fabrication in different conditions, in addition to our experimental studies of IR and R-FeNi ribbons and their MI modeling, we also investigated the ribbons with different FeNi coating thicknesses. The results are shown in Figure 5. It can be seen that the larger the MI, the smaller the thickness of the magnetic coating layer. Improvement in the MI effect is possible for a deposition of the FeNi layer of 10 nm. In this case, rather satisfactory sensitivity with respect to the applied magnetic field of 25 Ohm/Oe was obtained.

Note that the obtained dependence of MI on the coating thickness can also be used for the development of the sensors for reactive chemical agents that cause coating dissolution (destructive detection scheme) [52].

As the multicomponent detection includes various requirements, we also simulated MI of the ribbons with such types of coatings as Cu, Au, Fe, and Fe_3_O_4_. Earlier it was shown that MI elements based on amorphous ribbons can be effectively used as a part of the biosensor prototype to detect functional properties of biofluids [52] confirming the idea that MI CoFeSiB amorphous ribbon-based elements can be used in chemical sensors. However, the goal of the present study is different—to obtain a stable MI response in chemically active biological medium for possible detection of magnetically labeled biocomponents of interest. Therefore, CoFeCrSiB ribbons were employed to insure this stability.

The results of our studies suggest that the deposition of a thin-film coating with high magnetic permeability or high electrical conductivity on an amorphous ribbon can contribute to a significant increase in the sensitivity of sensors for aggressive chemical media based on MI. At the same time, it is necessary for detected medium to dissolve the thin-film coating. Figure 6a shows the magnetic field dependences of the impedance of the composite on the basis of amorphous ribbon obtained for coating thickness of 1000 nm. It can be seen that the most significant decrease in MI is caused by coatings with high conductivity (Au, Cu) and high magnetic permeability (Fe, FeNi) (see Table 1). In the case of the ribbons coated with Fe_3_O_4_, the high MI is retained (Figure 6b).

As noted in the introduction, the use of gold is considered in the development of biosensors of very different types, including magnetic biosensors based on MI as the gold coverings serve to create a self-assembling layers. The presented results of the modeling the effect of Au coating on MI can be useful for these purposes.

It is also important to emphasize that there is a large difference between the MI responses of Fe-coated ribbons and Fe_3_O_4_-coated ribbons because their maximum impedance differs by a factor of at least 6. As it is well known [56,57], the Fe_3_O_4_ layer is easily formed during the oxidation of Fe. Therefore, one can use the amorphous ribbon with a sufficient amount of iron in the composition or obtained required cover layer on the basis of Fe-coated ribbons, or depositthe Fe_3_O_4_ layer by sputtering technique or laser evaporation. In addition, one can compare the results of modeling with some experimental data available in the literature. The first attempt to obtain ribbon/film composite was made by Cerdeira et al. [33]. They reported on the experimental study of the CoFeMoSiB amorphous ribbons with the iron covering in the thickness interval of 0 to 240 nm. To some extent, the properties of CoFeCrSiB and CoFeMoSiB ribbons are similar and therefore comparison is valid and useful. However, the analysis of all obtained results indicates that lower thicknesses can be more effective, but the surface roughness can play very important role in contributing to the non-uniformity of the anisotropy of the amorphous ribbon-based composites. An additional iron layer can cause a modification of the roughness of the surface by a decrease of the depth of the Fe pores (see also Figure 3a).

As mentioned above, experimental studies have been undertaken into ribbon/film based composites. Ref. [58] reports a comparative study of uncoated (Fe50Ni50)81Nb7B12 nanocrystalline ribbons and ribbons coated with 120 nm of Co on both surfaces. The impact of the Co coating on the high frequency impedance of the (Fe50Ni50)81Nb7B12 ribbon was studied with techniques sensitive to surface magnetism. However, the thickness dependence was not discussed. We therefore propose to use the advantages of modeling techniques to make part of the studies via development of the appropriate models based on some experimental data.

In ref. [59] the authors studied copper oxide (CuO) film covering role on a surface of Co-based amorphous ribbon deposited using chemical successive ionic layer adsorption and reaction technique. The results showed that Co-based amorphous ribbons, which are coated CuO film, have a significant effect on the value and operation frequency for the MI effect as compared to the samples without coating. However, the overall MI ratio value was not very high.

In ref. [60] the authors describe a 50 nm-thick Co film grown either on the free surface or on the wheel-side surface of Co84.55Fe4.45Zr7B4 amorphous ribbons. They showed that the presence of the Co coating layer enhances both the MI ratio and its sensitivity. However, the maximum MI ratio was below 24%. This topic become very popular in recent years, and areas of interest include ideas for enhancing the GMI effect with both 3D metal and dielectric coatings [59,60,61,62]. However, this work is the first attempt to propose a methodology of the development of such a composite using modelization advantages which can be useful for the analysis of the experimental data of different authors.

## 4. Conclusions

The magnetoimpedance effect was experimentally investigated in amorphous Co_67_Fe_3_Cr_3_Si_15_B_12_ ribbons, both coated with a 1 μm thick Fe_20_Ni_80_ layer and without coating. The experimental studies were combined with computer FEM modeling in the range of the alternating current frequencies of 0.1 to 70 MHz.

It was found experimentally that the presence of the FeNi layer leads to a significant decrease in MI and a decrease in its magnetic field sensitivity. Thus, in the case of uncoated ribbons, the sensitivity reaches 5 Ohm/Oe at the AC frequency of 20 MHz, while in the case of FeNi-coated ribbons, the maximum sensitivity is 0.3 Ohm/Oe at the frequency of 30 MHz. The significant decrease in MI and the decrease in its magnetic field sensitivity after coating is associated with a decrease in the transverse effective magnetic permeability due to its “transcritical” state. The “transcritical” state of the FeNi coating was demonstrated using magneto-optical Kerr effect studies.

During computer simulation, it was possible to reproduce the main experimental result—the significant decrease in MI in FeNi-coated ribbons. In addition, it was shown that the simulated magnetoimpedance dependences are very close to the experimental ones at AC frequencies below 10 MHz for the uncoated ribbons and at the frequencies of 1 MHz and below for FeNi-coated ribbons. It was found that the discrepancy between the simulated and experimental dependences increases with an increase in the frequency of the alternating current. We proposed various methods to refine the computer model to achieve a better agreement between the simulated and experimental dependences in a wider AC frequencies range, taking into account the frequency dependence of the magnetic permeability, anisotropy dispersion, and the magnetic interaction of the coating and ribbon.

In addition, models were developed for the ribbons with different thickness of Fe_20_Ni_80_ coating and for the ribbons with such coatings as: Cu, Au, Fe, Fe_3_O_4_.

Some of the simulation results can find application in the creation of chemical sensors. Thus, the discovered strong dependence of MI on the coating thickness can be used in the development of sensors for chemical agents that release the coating. At the same time, the large difference in the MI response of Fe-coated and Fe_3_O_4_-coated ribbons can be used as the basis for sensors of chemicals that oxidize iron. It was also noted that the results of modeling Au coated ribbons can be useful in the design of magnetic biosensors.

## Figures and Tables

**Figure 1 sensors-21-06728-f001:**
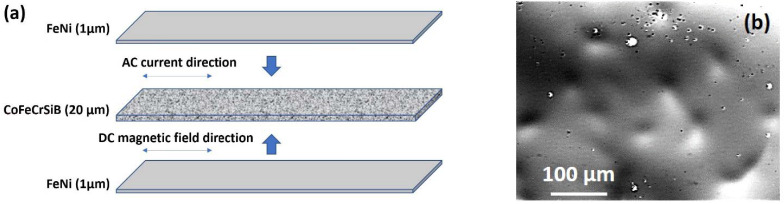
(**a**) Geometry of the model structure of FeNi/CoFeCrSiB/FeNi type. (**b**) Optical microscopy of the free surface of the amorphous CoFeCrSiB ribbon.

**Figure 2 sensors-21-06728-f002:**
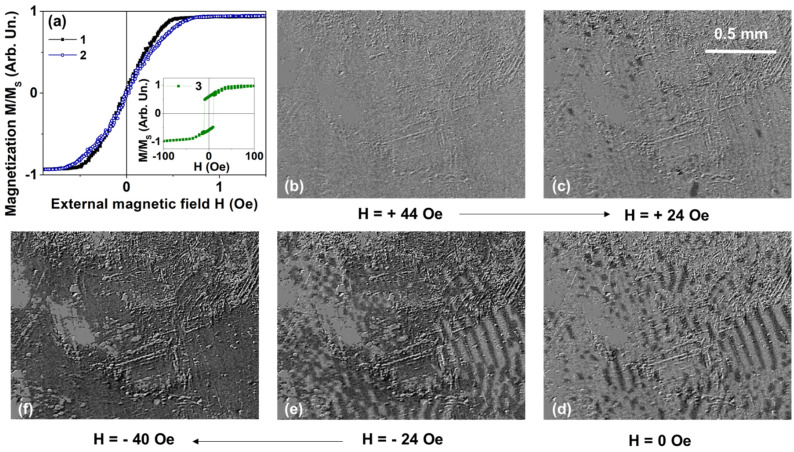
(**a**) Magnetoinductive magnetic hysteresis loops for the magnetic field applied in-plane of the ribbon and along the long side of it: (1)—the IR sample; (2)—the R-FeNi sample; and MOKE hysteresis loop (3)—Fe_20_Ni_80_ film 1 μm thick. (**b**–**f**) Magnetic domains of Co_67_Fe_3_Cr_3_Si_15_B_12_ IR-samples starting from magnetic saturation state in the high positive magnetic field.

**Figure 3 sensors-21-06728-f003:**
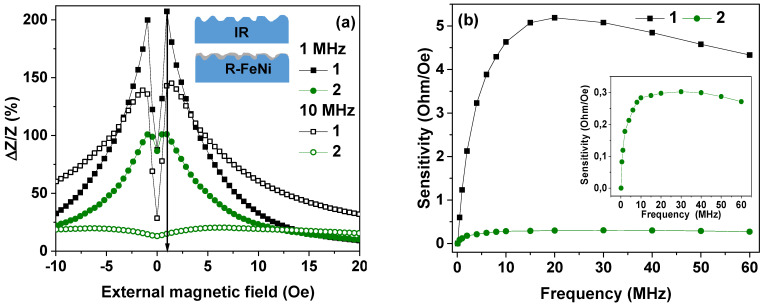
(**a**) Dependencies of the magnetoimpedance ratio on the value of the external magnetic field for different frequencies for the sample IR (black symbols, 1) and for the sample R-FeNi (green symbols, 2). The inset shows the description of the cross section of the IR ribbon and of the ribbon-based R-FeNi composite (not in the real scale) (**b**) Sensitivity dependencies of the MI ratio changes.

**Figure 4 sensors-21-06728-f004:**
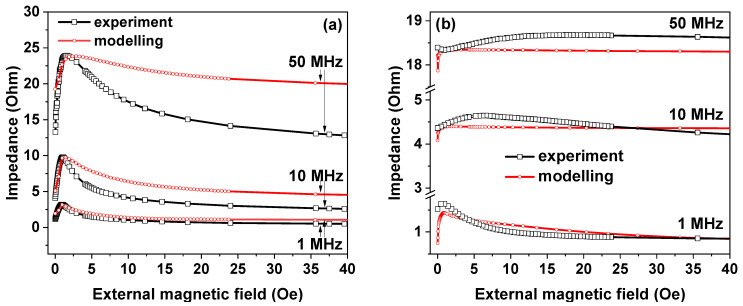
Difference between the experimental field dependence of the total impedance value and the same parameter obtained by the computer simulation: (**a**) for the IR (**b**) for the R-FeNi samples with a thickness of FeNi equal to 1 μm.

**Figure 5 sensors-21-06728-f005:**
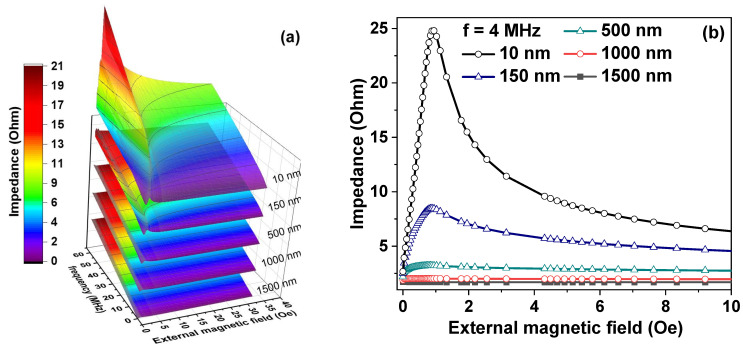
(**a**) Computing simulation of the dependence of the impedance modulus on the frequency and external magnetic field for different thickness of FeNi in the composite structure R-FeNi. (**b**) Computing modelling of absolute value of magnetoimpedance for frequency 4 MHz for different thicknesses of FeNi in the composite structure R-FeNi.

**Figure 6 sensors-21-06728-f006:**
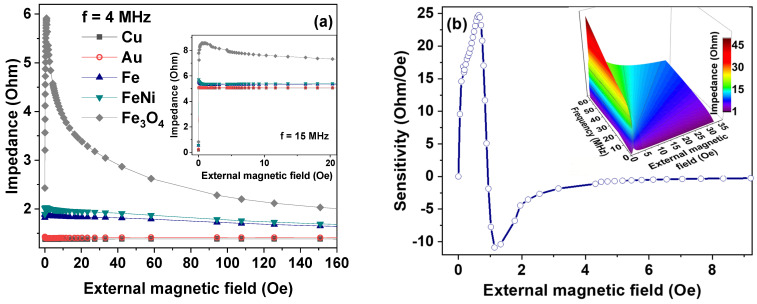
(**a**) Computer simulation of the field dependence of the impedance modulus at 4 MHz for the R-Cu, R-Au, R-Fe, R-FeNi and R-FeO samples (see Table 1) with the same covering thickness of the upper and lower layers of 1000 nm. The inset shows data for excitation current frequency *f* = 15 MHz. (**b**) Computer simulation of maximum sensitivity for structure R-FeO with thickness of upper and lower layers of 10 nm on 20 MHz. The inset shows dependence of absolute impedance value on frequency and external magnetic field values for this structure.

**Table 1 sensors-21-06728-t001:** Designation of the models (samples) of the CoFeCrSiB ribbons with various coatings with thickness 1 µm and some properties of coatings.

Designation of the Samples for Modelling	Structure	Covering	Magnetic Permeability (Calculated)	Electric Conductivity of Covering, S^−1^ (Constant)
IR	CoFeCrSiB	Absent	400–190,000	1.18 × 10^16^
R-FeNi	Fe_20_Ni_80_/CoFeCrSiB/Fe_20_Ni_80_	Fe_20_Ni_80_	400–190,000	7.65 × 10^17^
R-Cu	Cu/CoFeCrSiB/Cu	Cu	400–190,000	5.36 × 10^15^
R-Au	Au/CoFeCrSiB/Au	Au	400–190,000	3.69× 10^15^
R-Fe	Fe/CoFeCrSiB/Fe	Fe	200–120,000	1.01 × 10^7^
R-FeO	Fe_3_O_4_/CoFeCrSiB/Fe_3_O_4_	Fe_3_O_4_	1–2500	1.83 × 10^17^

## Data Availability

Data available from the corresponding author upon reasonable request.

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
