# Peer review of "Magnetoimpedance of CoFeCrSiB Ribbon-Based Sensitive Element with FeNi Covering: Experiment and Modeling"

_sensors, 2021, doi:10.3390/s21206728_

Round 1

Reviewer 1 Report

This revised manuscript has completed the reviewer's comments of the previous manuscript.  It is thus suitable for publishing in Sensors. 

Author Response

We are grateful to the reviewer for the appreciation of our work.

Reviewer 2 Report

The paper «Magnetoimpedance of CoFeCrSiB ribbon based sensitive element with FeNi covering: experimental and modeling» presented by Stanislav Volchkov and coworkers, represents an experimental and computational study of sensitive properties of magnetic ribbons. Authors fabricated ribbon/film composite and found experimentally that the presence of the FeNi film leads to a significant decrease in magnetoimpedance (MI) and a decrease in its magnetic field sensitivity. MI calculations well-described experimental data were done for the ribbons with different thicknesses of FeNi coating and for the ribbons with such coatings as Cu, Au, Fe, Fe3O4. From the calculation was found that the thin film coatings with high magnetic permeability or high electrical conductivity on an amorphous ribbon could increase the sensitivity of the sensor based on MI. The article is well written, structured, and organized. Presented results contain useful information for readers who are interested in MI effect based sensors. In my opinion, the article satisfies the requirements of Sensors Journal. I recommend to accept the publication as is.

Author Response

(The authors gave the same response as above.)

Reviewer 3 Report

This has nice comparison between simulation and experiment, but it could be strengthened by comparing separately comparing the real and imaginary parts of the simulated and experimental impedance (Fig. 4). 

Also please explain how the permeability was calculated in Table 1 in addition to the conductivity being quoted to 3 significant digits when the temperature is not given. 

There is ambiguity between θ representing the angle between the magnetic field and magnetization and that representing the diffraction angle. 

Finally. please explain the choice of ψ was determined and how it reflected the symmetry of the growth process.

Author Response

Dear Reviewer,

We would like to submit revised version of the manuscript “Magnetoimpedance of CoFeCrSiB ribbon based sensitive element with FeNi covering: experimental and modeling” Stanislav O. Volchkov, Anna A. Pasynkova, Michael S. Derevyanko, Dmitry A. Bukreev, Nikita V. Kozlov, Andrey V. Svalov, Alexander V. Semirov for MDPI Sensors, SI "Sensors and Biosensors Related to Magnetic Nanoparticles" for the fourth round of the review and possible publication. All Reviewers suggestions were taken into account and we submit revised text in which all changes are outlined.

Point 1:  This has nice comparison between simulation and experiment, but it could be strengthened by comparing separately comparing the real and imaginary parts of the simulated and experimental impedance (Fig. 4).

Response 1:

Undoubtedly, the division into active and reactive parts of impedance can more clearly show research processes. So, for example, it was shown in research [doi.org/10.1016/S1567-2719(03)15005-6], including our previous works [doi.org/10.1088/0256-307X/24/5/064] that the curve of active resistance ratio has a sharper peak and stronger impedance's hysteresis than the curves describing the reactance and full impedance behavior. DR/R shows a maximum variation value and maximum sensitivity with respect to the applied field, calculated as a change per unit of the external field (of order of 420% in units of A/m for ΔR/R compared to with 210% for ΔZ/Z response for initial state). And this higher sensitivity of the resistance can be used for the design of the detectors working on the principle of the separation of the impedance components. But in this work, no significant difference was revealed when the active and reactive parts show global changes. This can be seen in Fig. 1 Therefore, the description of the impedance was chosen in principle.

(see attachment)

Fig. 1. The dependences of the relative changes in the impedance modulus ΔZ/Z, its real ΔZ’/Z’ and imaginary ΔZ”/Z” components on a constant external magnetic field for 4 MHz.

Point 2:  Also please explain how the permeability was calculated in Table 1 in addition to the conductivity being quoted to 3 significant digits when the temperature is not given.

Response 2:

The magnetic permeability was calculated based on formula (4). We are deeply grateful to the referee for comments on the accuracy of the given values. Therefore, we made changes to the table taking into account the possible calculation error.

Point 3:  There is ambiguity between θ representing the angle between the magnetic field and magnetization and that representing the diffraction angle.

Response 3:

Thanks for your comment, a corresponding change has been made to one of the references.

Point 4:  Finally. please explain the choice of ψ was determined and how it reflected the symmetry of the growth process.

Response 4:

The choice of the angle of anisotropy was based on past parametric studies [20, 43] in such a way that the maximum magnetic permeability of a sample with a transverse uniaxial magnetic anisotropy was achieved at a nonzero value of the linear magnetization component. Reference [43] was given in Section 2.2 dedicated to the description of modeling, where the selection of the mentioned parameters was discussed in detail. The choice of the ψ was based on work [20].

Best regards,

Dr. Anna A. Pasynkova

Reviewer 4 Report

The manuscript entitled “Magnetoimpedance of CoFeCrSiB ribbon based sensitive element with FeNi covering experimental and modeling” by Stanislav O. Volchkov et. al. mainly deals with the investigation of the magnetoimpedance of rapidly quenched Co67Fe3Cr3Si15B12 amorphous ribbons and magnetic Fe20Ni80/Co67Fe3Cr3Si15B12/Fe20Ni80 composites. The manuscript is well written and well organized. I have few comments: the authors should address the comments raised by the reviewer.

  1. The advantages of Co67Fe3Cr3Si15B12 and Fe20Ni80 should be discussed with suitable references in introduction.
  2. The authors should prove that the observed MI effect observed in this study is not due to artifacts. They should discuss by referring the paper “Magnetocapacitance without magnetoelectric coupling Appl. Phys. Lett. 88, 102902 (2006)”.
  3. The symmetric/asymmetric behavior of Fig.3 (a) should be discussed in detail in terms of nature of MI effect.
  4. The possible applications of this materials should be mentioned.
  5. The authors should provide some recent references and in few places references are missing.
  6. There are some sentences with grammar or typo problems; a careful reading is required.

Author Response

Dear Reviewer,

We would like to submit revised version of the manuscript “Magnetoimpedance of CoFeCrSiB ribbon based sensitive element with FeNi covering: experimental and modeling” Stanislav O. Volchkov, Anna A. Pasynkova, Michael S. Derevyanko, Dmitry A. Bukreev, Nikita V. Kozlov, Andrey V. Svalov, Alexander V. Semirov for MDPI Sensors, SI "Sensors and Biosensors Related to Magnetic Nanoparticles" for the fourth round of the review and possible publication. All Reviewers suggestions were taken into account and we submit revised text in which all changes are outlined.

Point 1:  The advantages of Co67Fe3Cr3Si15B12 and Fe20Ni80 should be discussed with suitable references in introduction.

Response 1:

Сobalt-based ribbons is one of well-known soft magnetic material with low coercivity, high magnetic permeability and saturation  magnetization [DOI: 10.1109/TMAG.1986.1064399]. Type CoFeSiB alloys with a prevailing cobalt content were discussed in the introduction in references [5-10] and [12, 16]. Soft magnetic properties of Co-based ribbons balanced so that zero magnetostriction is achieved. These properties lead to a high magnetoimpedance effect due to significant change in dynamic magnetic permeability. Cobalt-chromium ribbons are soft magnetic materials with high temperature stability of their properties as well as high initial magnetic permeability [ref. 17, 18 was added in the text].

Permalloy is also a familiar magnetic material showing high magnetoimpedance properties. The text provides links for a more detailed description of the advantages of permalloy in the field of prototypes of various types [27,28,32-33]. It is especially important that there are well established technologies for FeNi films deposition with very high degree of control of the composition and other properties.

Point 2:  The authors should prove that the observed MI effect observed in this study is not due to artifacts. They should discuss by referring the paper “Magnetocapacitance without magnetoelectric coupling Appl. Phys. Lett. 88, 102902 (2006)”.

Response 2:

We are grateful to the referee for pointing out this article about the features of the magnetocapacitance effect in ferroelectrics. Indeed, it is possible to draw a parallel between these effects, but they are opposite in physical basis. First, both amorphous cobalt-based alloy and permalloy are classical ferromagnets. Second, the giant magnetoimpedance effect is observed due to the skin effect, which depends on the change in the magnetic permeability of the ferromagnet in the presence of an external magnetic field. This article cites one of the first works devoted to the giant magnetoimpedance effect Beach, R.S.; Berkowitz, A.E. Giant magnetic field dependent impedance of amorphous FeCoSiB wire. Appl. Phys. Lett. 1994, 64, 3652–3654. It is comparable to the recommendation of a Reviewer on the magnetocapacitive effect in terms of the number of citations as one of the basic works in the field of magnetoimpedance research. Our study was devoted to the GMI effect with high sensitivity to external magnetic field in the range of several oersteds.

Point 3:  The symmetric/asymmetric behavior of Fig.3 (a) should be discussed in detail in terms of nature of MI effect.

Response 3:

On the one hand, the issue of peak asymmetry was discussed in the literature earlier (the ref. DOI: 10.4028/www.scientific.net/msf.302-303.209 was added to the text). According to the proposed theory, there is an induced perpendicular anisotropy of magnetoelastic origin and also a hysteresis of magnetization, which leads to a small divergence of the GMI peaks.

Point 4:  The possible applications of this materials should be mentioned.

Response 4:

We would like to attract attention to the Reviewer that to some extent the concept of the chemical sensing was discussed (related Ref. [53]). However, the best MI sensitive element for chemical version is the CoFeSiB element without chromium, which is different from the present case. The discovered strong dependence of MI on the coating thickness can be used in the development of sensors for chemical agents that release the coating.  The Sensors-1360653 manuscript does not focused on chemical sensors, discussed composite seems to be first investigation of such configuration and establishing its benefits.

Point 5:  The authors should provide some recent references and in few places references are missing.

Response 5:

The text has been revised according to the reviewer's notes.

Point 6:  There are some sentences with grammar or typo problems; a careful reading is required.

Response 6:

Thank you for your recommendation, we made additional English proof-reading in order to address your comment.

Best regards,

Dr. Anna A. Pasynkova

This manuscript is a resubmission of an earlier submission. The following is a list of the peer review reports and author responses from that submission.

Round 1

Reviewer 1 Report

Magnetoimpedance of CoFeCrSiB ribbon based sensitive element with FeNi covering: experimental and modeling

By S. O. Volchkov et al.

The authors deal with the magnetoimpedance effect of the amorphous CoFeCrSiB ribbon with and without FeNi film. Magnetic properties and magnetoimpedance were discussed based on the experimental data. Finite element analysis was also performed to examine the magnetoimpedance effect. This manuscript is timely. However, the reviewer would suggest the authors to address the following revisions:

  1. There is no explanation why the authors chose this CoFeCrSiB ribbon among the many magnetic materials. Similarly, why did the authors select the FeNi layer?
  2. Does the finite element model take material deformation into account?
  3. Page 5, line 214: Please check (g)-(h)-(i).
  4. Fig. 2 (a): It is difficult to understand the results because the character is too small. (3) in the caption is completely invisible.

Author Response

Dear Editor,

We would like to submit revised version of the manuscript “Magnetoimpedance of CoFeCrSiB ribbon based sensitive element with FeNi covering: experimental and modeling

Stanislav O. Volchkov, Anna A. Pasynkova, Michael S. Derevyanko3 Dmitry A. Bukreev, Nikita V. Kozlov, Andrey V. Svalov, Alexander V. Semirov for MDPI Sensors, SI “"Sensors and Biosensors Related to Magnetic Nanoparticles" for the second round of the review and possible publication. All Reviewers suggestions were taken into account and we submit revised text in which all changes are outlined.

Response to Reviewer 1 Comments:

Point 1: There is no explanation why the authors chose this CoFeCrSiB ribbon among the many magnetic materials. Similarly, why did the authors select the FeNi layer?

Response 1:

Сobalt-based ribbons is one of well-known soft magnetic material with low coercivity, high  magnetic permeability  and  saturation  magnetization. Soft magnetic properties of Co-based ribbons balanced so that zero magnetostriction is achieved. These properties lead to a high magnetoimpedance effect. Cobalt-chromium ribbons are soft magnetic materials with high temperature stability of their properties as well as high initial magnetic permeability. As noted in the text, these are the materials chosen for the prototype detection of the oxide of magnetic nanoparticles of iron oxide in commercial ferrofluid.

Permalloy is also a familiar magnetic material showing high magnetoimpedance properties. It is especially important that there are well established technologies for FeNi films deposition with very high degree of control of the composition and other properties.

The following text has been added:

  1. i) « … as a magnetic sensitive element [14,15], and have competitively high magnetoimpedance properties [doi.org/10.1016/j.jmmm.2006.02.124]»
  2. ii) «…MI effect [23]. Thus, permalloy obtained by magnetron sputtering is a good choice due to low deviation of the angle of magnetic anisotropy. … So, it seems expedient to use gold or Fe3O4 coatings widely adapted in the self-assembling processes in biotechnologies in the tasks of marker biodetection [4, 10].»

iii) « … the following coatings were developed: FeNi, Cu, Au, Fe, Fe3O4. Copper and gold based coatings have been selected as popular high conductivity materials. »

Point 2: Does the finite element model take material deformation into account?

Response 2:

This model does not take into account the deformation of materials, both for the cobalt-based layer and for the permalloy. The main reason for this is that both Co-based ribbons and FeNi films of the selected compositions have close to zero value of the magnetostriction constant. The authors suppose that the contribution from the deformation process, if such is observed, does not have a significant effect on effective transverse magnetic permeability, which can be obtained using the procedure for minimizing the free energy functional. This fact follows directly from the formula (4) for effective transverse magnetic permeability, which there are no parameters that significantly depend from the deformation of the materials. In addition, according to this work, the main contribution to the change in impedance is produced by a change in the geometry of materials of the order of hundreds of nanometres, which significantly exceeds the potential deformations.

Point 3: Page 5, line 214: Please check (g)-(h)-(i).

Response 3:

Thank you for your recommendation, the mistype has been corrected.

Point 4: Fig. 2 (a): It is difficult to understand the results because the character is too small. (3) in the caption is completely invisible.

Response 4:

Thank you for your recommendation, the fonts of Figure 2a have been increased.

Best regards,

Dr. Anna A. Pasynkova

Reviewer 2 Report

The Fe20Ni80 is deposited by sputtering, which is difficult to control its proportion of ingredients, so you need give a results thin films analyses.

In fact, this simulation data are not in agree with experiment results.

In the conclusion, your materials can be used as chemical sensors, however on results to verify it in the paper.

The whole paper give no many valuable conclusions

Author Response

We would like to submit revised version of the manuscript “Magnetoimpedance of CoFeCrSiB ribbon based sensitive element with FeNi covering: experimental and modeling

Stanislav O. Volchkov, Anna A. Pasynkova, Michael S. Derevyanko3 Dmitry A. Bukreev, Nikita V. Kozlov, Andrey V. Svalov, Alexander V. Semirov for MDPI Sensors, SI “Sensors and Biosensors Related to Magnetic Nanoparticles" for the second round of the review and possible publication. All Reviewers suggestions were taken into account and we submit revised text in which all changes are outlined.

Response to Reviewer 2 Comments

Thank you for sharing your opinion. We made additional English proof-reading in order to address your comment “Extensive editing of English language and style required”.

Point 1: The Fe20Ni80 is deposited by sputtering, which is difficult to control its proportion of ingredients, so you need give a results thin films analyses.

Response 1:

Permalloy layers obtained by magnetron sputtering have been repeatedly investigated  by scientific community and in special conditions the composition can be very well controlled  using modern deposition techniques:

Wang et al. High resonance frequencies induced by in-plane antiparallel magnetization in NiFe/FeMn bilayer. J. Magn. Magn.Mater. 514 (2020) 167139. https://doi.org/10.1016/j.jmmm.2020.167139

Yang et al. The abnormal damping behavior due to the combination between spin pumping and spin back flow in Ni80Fe20/Rut bilayers. J. Magn. Magn.Mater. 502 (2020) 166495. https://doi.org/10.1016/j.jmmm.2020.166495

Li et al. Magnetic properties of permalloy films with different thicknesses deposited onto obliquely sputtered Cu under layers. J. Magn. Magn.Mater. 377 (2015) 142. http://dx.doi.org/10.1016/j.jmmm.2014.10.029

Svalov et al. Structure and magnetic properties of thin permalloy films near the “transcritical” state. IEEE Trans. Magn. 46 (2010) 333. http://dx.doi.org/10.1109/TMAG.2009.2032519

Indeed, every time the deposition is made we add additional piece of the substrate in the area of uniform deposition and the composition is checked afterwards either by EDX technique and/or by TXRF measurements carried out by a Nanohunter spectrometer. One of the co-authors of submitted manuscript (Dr. A.V. Svalov) has 30 years experienced in thin film deposition proven by the WOK record all depositions were made with certified equipment. We add requested comment to the text.

Point 2: In fact, this simulation data are not in agree with experiment results.

Response 2:

The specified difference between experimental and theoretical results was noted. At the current stage, we proposed the explanation of this discrepancy, and also outlined the direction of further research.

The key result of this article was to achieve a qualitative agreement between simulation and experiment. In the works presented earlier, the error could reach 10% [F.L.A. Machado, S.M. Rezende / Journal of Applied Physics 79(8, PART 2B) (1996) 6558 – 6560]. The analytical model presented earlier had a comparable margin of error [N.A. Buznikov et.al. / Biosensors and Bioelectronics 117 (2018) 366 – 372]. Some minor fitting had to be made in order to adapt the model to amorphous ribbons (the films based on the Fe-Ni was the object of the study). This analytical model was the basis of the Сomsol software model in this work and it had a close margin of error.

Point 3: In the conclusion, your materials can be used as chemical sensors, however on results to verify it in the paper.

Response 3:

The Sensors-1309441 manuscript does not focused on chemical sensors, discussed composite seems to be first investigation of such configuration and establishing its benefits. We add some discussion to the text, including part related to the chemical sensors.

We would like to attract attention to the Reviewer that to some extent the concept of the chemical sensing was discussed in the first version of the manuscript (related Ref. Kurlyandskaya; G.V.; Fal Miyar, V. Surface modified amorphous ribbon based magnetoimpedance biosensor. Biosensors and Bioelectronics 2007, 22(9-10), 2341–2345). However, the best MI sensitive element for chemical version is the CoFeSiB element without chromium, which is different from the present case.

Point 4: The whole paper give no many valuable conclusions.

Response 4:

Dear reviewer, we really appreciate your opinion. One of the main tasks of MI-sensors development is optimization of their magnetoimpedance parameters in accordance with particular application request especially biological one. This work presents the results of experimental and modelling studies of the magnetoimpedance effect of the amorphous CoFeCrSiB ribbons both with and without FeNi magnetic layer coating (Cu, Au, Fe, Fe3O4), which could be used to functionalize their surface in real biomedical devices or chemical sensors. Our result demonstrates that the deposition of specialized coatings on amorphous ribbons can be not only a way to biofunctionalize the sensitive element surface, but also increase the magnetoimpedance characteristics. In our opinion, this study will be of broad interest to those investigating in low magnetic field sensors. For example, Yijun Chen, Jintang, Zoua, Xiangfeng Shua, Yenan Songa and Zhenjie Zhao discussed the  enhancement of giant magneto-impedance effect in sandwich FINEMET/rGO/FeCo composite ribbons (Applied Surface Science 545, 2021, 149021) which would be much easier to develop in a future using the model, proposed in the present study.

Best regards,

Dr. Anna A. Pasynkova

Reviewer 3 Report

The paper concerns experimental and numerical studies of the FEM type of magnetoimpedance of CoFeCrSiB ribbon with FeNi covering. Presented research problem and its solution makes an important contribution to the consideration of this publication in this journal. The paper presents a very well-developed form for understanding relevant references in this field, and the range of cited literature sources is extensive. The paper’s argument is built on an appropriate base of theory and ideas. In this case, the applied research methods and tools are clear and legible. The results were analysed in detail giving appropriate conclusions. The presented research results make a significant contribution to the development of science. In my opinion, the paper enriches the research area related to the search for the magnetoimpedance responses of amorphous ribbons for the estimation of their properties.

Minor remarks concern some vague statements, namely:

  1. It is advisable to show the parameters adopted in the modeling (MS, HK, σ and ψ).
  2. What was the criterion for selecting the FeNi covering thickness (see Fig. 5b)?
  3. Referring to the comments on the results presented in Fig. 6a, whether similar conclusions can be made for covering of other thicknesses (see Fig. 5).
  4. Figure 6b needs some comment.

The remaining comments relate to minor editorials ,especially as that Figure have illegible descriptions in their structure (see Fig. 2a or Fig. 6b). The use terminology is correct. It does not raise any objections.

Finally, I recommend this paper for publication after minor revision.

Author Response

Dear Editor anf Reviewer,

We would like to submit revised version of the manuscript “Magnetoimpedance of CoFeCrSiB ribbon based sensitive element with FeNi covering: experimental and modeling

Stanislav O. Volchkov, Anna A. Pasynkova, Michael S. Derevyanko3 Dmitry A. Bukreev, Nikita V. Kozlov, Andrey V. Svalov, Alexander V. Semirov for MDPI Sensors, SI "Sensors and Biosensors Related to Magnetic Nanoparticles" for the second round of the review and possible publication. All Reviewers suggestions were taken into account and we submit revised text in which all changes are outlined.

Response to Reviewer 3 Comments

Point 1: It is advisable to show the parameters adopted in the modeling (MS, HK, σ and ψ).

Response 1: Thank you for your recommendation, information has been included into the text.

Point 2: What was the criterion for selecting the FeNi covering thickness (see Fig. 5b)?

Response 2: The selection criteria for the thickness in Figure 5b are based on the best visualization of the impedance decreasing trend from 10 nm to 1500 nm. The thickness of 1500 nm is quite limiting for real deposition technology. It is assumed that intermediate thicknesses between 10 nm and 150 nm fit into the overall dynamics of Figure 5b. We also add the comment on the experimental study of the CoFeMoSiB amorphous ribbons with the iron covering where they studied the thicknesses of 0 to 240 nm (Cerdeira, M.A.; Kurlyandskaya, G.V.; Fernandez, A.; Tejedor, M.; Garcia-Miquel, H. Giant magnetoimpedance effect in surface modified CoFeMoSiB amorphous ribbons. Chin. Phys. Lett. 2003 20(12), 2246-2249).

Point 3: Referring to the comments on the results presented in Fig. 6a, whether similar conclusions can be made for covering of other thicknesses (see Fig. 5).

Response 3: If we understood the reviewer correctly, it was proposed to consider the MI characteristics depending on the thickness not only for permalloy, but also for other coating compositions. In our opinion, the studied coatings 1000 nm thick differ significantly in their electromagnetic properties. Reducing the thickness can have a complex effect on the MI characteristics of composites. Therefore, we believe that additional calculations and simulations are necessary in order to draw a conclusion about the thickness dependences for coatings of other compositions.

Point 4: Figure 6b needs some comment.

Response 4:  The largest change in the MI effect at the corresponding frequencies was discussed using the example of Fig. 6a. The inset in Figure 6b shows the well-known MI curve with a maximum in low fields. The frequency dependence does not reach its maximum in the studied range, which may be a consequence of the model simplifications specified in the article. This issue requires further research and experimental reinforcement. Therefore, in this figure, we have focused on proving the field dependence of the sensitivity for the composite model with the best MI characteristics. A corresponding reference to the figure has been added to the text.

Point 5: Figure have illegible descriptions in their structure (see Fig. 2a or Fig. 6b)

Response 5: Thank you for your recommendation, the fonts of Figure 2a and Figure 6b have been increased.

Best regards,

Dr. Anna A. Pasynkova

Reviewer 4 Report

Referee Report

 “Magnetoimpedance of CoFeCrSiB ribbon based sensitive element with FeNi covering: experimental and modeling”

by authors  Stanislav O. Volchkov, Anna A. Pasynkova, Michael S. Derevyanko, Dmitry A. Bukreev, Nikita V. Kozlov, Andrey V. Svalov, Alexander V. Semirov

submitted to Sensors

This article presents the results of studies of the magnetoimpedance effect of the amorphous CoFeCrSiB ribbons and ribbons with FeNi coating  The experimental data are compared with the simulation results obtained using finite element method. The simulation and experimental results are in good agreement and confirm each other. The article is well constructed and interesting to read. Obtained results can be useful for design of MI sensors development including MI biosensors for magnetic label detection. I believe the article has good potential, despite the comments I have left below.

However, minor revision is required before the article can be published in Sensors.

Comments

  • English needs to be improved.

  • The motivation for using a NiFe layer on the surface of the CoFeCrSiB ribbon is not explained. The choice of the coating was due to the optimal magnetic characteristics of the material? Would such a structure be suitable for the bioapplications discussed in the article? Poor corrosion resistance of permalloy can also be a problem.

  • Magnetron sputtering has good covering capacity, therefore care must be taken to ensure that the ends of the ribbon (multilayer structure) are not coated be NiFe. The end-face coating can affect the magnetoimpedance effect. Was this controlled in this work? Why was magnetron sputtering chosen for the coating deposition? For example, the method of electrodeposition is cheaper and allows you to control the parameters of the microstructure of the coatings, and hence the magnetic properties. Perhaps the authors will be interested in the method described in the article [1]
  1. Anomalies in Ni-Fe nanogranular films growth T.I. Zubar et al. Journal of Alloys and Compounds 748 (2018) 970-978 https://doi.org/10.1016/j.jallcom.2018.03.245

  • In general, the procedure for synthesizing samples is presented too briefly.

  • Much attention is paid to the domain structure of samples, which was studied using the Kerr effect. It is obvious that there is a strong relationship between the domain structure and the microstructure of the material. For example, it was shown in [2] that fundamentally different domain structures can be formed in NiFe films, depending on the structure. I want to note the lack of results of the study of the crystal structure and microstructure as a disadvantage. It can probably be corrected by providing references to previous works that contain relevant information.
  1. Tatiana Zubar et al. Features of the Growth Processes and Magnetic Domain Structure of NiFe Nano-objects J. Phys. Chem. C 2019, 123, 26957−26964. DOI: 10.1021/acs.jpcc.9b06997

  • The results showed that FeNi coating does not increase the magnetoimpedance effect. The simulation results provided an explanation for this. The scientific and practical significance of the study is sufficient without additions, however, I recommend giving some rough estimates for the further development of the topic. Are there ways to improve the MI effect by NiFe coatings and functionalizing the surface? Is it possible to vary the parameters of the structure of the deposited layer, or is the use of layers with magnetic anisotropy capable of improving the MI effect?

  • The resolution of Figure 2 is too bad and does not allow to see the details on the graph (2, a).

Author Response

We would like to submit revised version of the manuscript “Magnetoimpedance of CoFeCrSiB ribbon based sensitive element with FeNi covering: experimental and modeling

Stanislav O. Volchkov, Anna A. Pasynkova, Michael S. Derevyanko3 Dmitry A. Bukreev, Nikita V. Kozlov, Andrey V. Svalov, Alexander V. Semirov for MDPI Sensors, SI "Sensors and Biosensors Related to Magnetic Nanoparticles" for the second round of the review and possible publication. All Reviewers suggestions were taken into account and we submit revised text in which all changes are outlined.

Response to Reviewer 4 Comments

Point 1: English needs to be improved.

Response 1: Thank you for sharing your opinion. We made additional English proof-reading in order to address your comment “English language and style are fine/minor spell check required”.

Point 2: The motivation for using a NiFe layer on the surface of the CoFeCrSiB ribbon is not explained. The choice of the coating was due to the optimal magnetic characteristics of the material? Would such a structure be suitable for the bioapplications discussed in the article? Poor corrosion resistance of permalloy can also be a problem.

Response 2: Permalloy is a well-known material showing high magnetoimpedance properties. In this context, it is important that permalloy materials were investigated in a multilayered configuration as noted on page 2. We made requested changes in the text.

Point 3: Magnetron sputtering has good covering capacity, therefore care must be taken to ensure that the ends of the ribbon (multilayer structure) are not coated be NiFe. The end-face coating can affect the magnetoimpedance effect. Was this controlled in this work? Why was magnetron sputtering chosen for the coating deposition? For example, the method of electrodeposition is cheaper and allows you to control the parameters of the microstructure of the coatings, and hence the magnetic properties. Perhaps the authors will be interested in the method described in the article [1]

[1] Anomalies in Ni-Fe nanogranular films growth T.I. Zubar et al. Journal of Alloys and Compounds 748 (2018) 970-978 https://doi.org/10.1016/j.jallcom.2018.03.245

In general, the procedure for synthesizing samples is presented too briefly.

Response 3: The ends of the ribbon were free from the permalloy coating and, after obtaining the multilayer composite, were cut off to obtain a sample of the length required for GMI measurements. The choice of magnetron sputtering was motivated by the high stability of this technique and the high coating production rate. It was interesting for us to get acquainted with T.I. Zubar and co-author’s work devoted to the study of electrodeposited permalloy films. In this article, it was demonstrated that a small size distribution of nanograins is observed for thick 800 nm permalloy films. In this regard, we can assume a lower dispersion of the easy axes, which was shown earlier in a large number of studies [A. Hubert; R. Schäfer, Magnetic Domains. Springer: Berlin, Germany, 1998, p. 423-425; A. V. Svalov  et. al. / IEEE Transactions on Magnetics, 46(2) (2010) 333–336. doi:10.1109/tmag.2009.2032519; J.-M. Quemper et. al. / Sensors and Actuators A: Physical, 74(1-3), (1999) 1–4. doi:10.1016/s0924-4247(98)00323-9].

On the one hand, magnetron sputtering is a well-known method [McLeod PS, Hartsough LD. J Vac Sci Technol 1977;14(1):263-265; P. Kelly,, R. Arnell, Vacuum, 56(3) (2000) 159–172. https://doi.org/10.1016/s0042-207x(99)00189-x] and the conditions required to obtain a stable layer (eg permalloy), which to some extent differ depending on the installation. The following parameters were applied for the mentioned equipment: the background pressure was 3.0 ´10-7 mbar and working argon pressure 3.8´10-3 mbar.

Point 4: Much attention is paid to the domain structure of samples, which was studied using the Kerr effect. It is obvious that there is a strong relationship between the domain structure and the microstructure of the material. For example, it was shown in [2] that fundamentally different domain structures can be formed in NiFe films, depending on the structure. I want to note the lack of results of the study of the crystal structure and microstructure as a disadvantage. It can probably be corrected by providing references to previous works that contain relevant information.

Tatiana Zubar et al. Features of the Growth Processes and Magnetic Domain Structure of NiFe Nano-objects J. Phys. Chem. C 2019, 123, 26957−26964. DOI:10.1021/acs.jpcc.9b06997

Response 4:

The main attention in the work of the authors is devoted to amorphous ribbons and FeNi film coatings with a thickness of up to 1 μm. Magnetic nanoparticles, even as features of the topology of films with a thickness of up to 50 nm, exhibit specific features of the magnetic structure, which are less marked in 20 times thicker films.

MOKE probes samples over a depth which is the penetration depth of light. The penetration depth is about 20 nm in amorphous ribbons. In this regard, the work presented the results for Co-base ribbons and for nanocrystalline films with 1 μm thick permalloy (in the transcritical state).

Eventually the near-surface features of the experimentally investigated FeNi film coating of sandwich had a small contribution to the magnetostatic and magnetodynamic properties. In addition to that, the main magnetic characteristics of permalloy film layers have already been studied earlier [A. Hubert; R. Schäfer, Magnetic Domains. Springer: Berlin, Germany, 1998, p. 423-425; A. V. Svalov et. al. / IEEE Transactions on Magnetics, 46(2) (2010) 333–336. doi:10.1109/tmag.2009.2032519].

Point 5: The results showed that FeNi coating does not increase the magnetoimpedance effect. The simulation results provided an explanation for this. The scientific and practical significance of the study is sufficient without additions, however, I recommend giving some rough estimates for the further development of the topic. Are there ways to improve the MI effect by NiFe coatings and functionalizing the surface? Is it possible to vary the parameters of the structure of the deposited layer, or is the use of layers with magnetic anisotropy capable of improving the MI effect?

Response 5: In this work, it was shown by model that a significant improvement in the magnetoimpedance effect is possible for a deposition layer FeNi of 10 nm. In this case, the sensitivity reaches 25 Ohm/Oe and there are no processes associated with unwanted thickness effects that affect on the effective magnetic permeability.

Point 6: The resolution of Figure 2 is too bad and does not allow to see the details on the graph (2, a).

Response 6: Thank you for your recommendation, the fonts of Figure 2a have been increased.

Best regards,

Dr. Anna A. Pasynkova